# Compounds from the Petroleum Ether Extract of *Wedelia chinensis* with Cytotoxic, Anticholinesterase, Antioxidant, and Antimicrobial Activities

**DOI:** 10.3390/molecules28020793

**Published:** 2023-01-13

**Authors:** Md. Abdur Rashid Khan, Md. Aminul Islam, Kushal Biswas, Md. Yusuf Al-Amin, Md. Salim Ahammed, Md. Imran Nur Manik, KM Monirul Islam, Md. Abdul Kader, AHM Khurshid Alam, Shahed Zaman, Golam Sadik

**Affiliations:** 1Department of Pharmacy, University of Rajshahi, Rajshahi 6205, Bangladesh; 2Department of Chemistry, University of Rajshahi, Rajshahi 6205, Bangladesh; 3Department of Biomedical and Nutritional Sciences, Zuckerberg College of Health Sciences, University of Massachusetts Lowell, Lowell, MA 01854, USA; 4Purdue University Interdisciplinary Life Sciences Graduate Program, Purdue University, West Lafayette, IN 47907, USA; 5Basic Biomedical Sciences, Sanford School of Medicine, University of South Dakota, Vermillion, SD 57069, USA; 6Department of Bioscience, Graduate School of Science and Technology, National University Corporation Shizuoka University, 836 Ohyo, Suruga-ku, Shizuoka 422-8529, Japan

**Keywords:** medicinal plants, kaurene diterpenoids, steroids, bioactivities, chronic diseases

## Abstract

*Wedelia chinensis* is a folk medicine used in many Asian countries to treat various ailments. Earlier investigations reported that the petroleum ether extract of the plant has potential biological activity, but the compounds responsible for activity are not yet completely known. Therefore, the current work was designed to isolate and characterize the compounds from the petroleum ether extract and to study their bioactivities. Four compounds including two diterepenes (-) kaur-16α-hydroxy-19-oic acid (**1**) and (-) kaur-16-en-19-oic acid (**2**), and two steroids β-sitosterol (**3**), and cholesta-5,23-dien-3-ol (**4**) were isolated and characterized. Among the compounds, the diterpenes were found to have more biological activities than the steroidal compounds. Compound **1** showed the highest cytotoxicity with LC_50_ of 12.42 ± 0.87 μg/mL. Likewise, it possesses good antioxidant activity in terms of reducing power. On the contrary, compound **2** exerted the highest antiacetylcholinesterase and antibutyrylcholinesterase activity. Both the diterpenes showed almost similar antibacterial and antifungal activity. The identification of diterpenoid and steroid compounds with multifunctional activities suggests that *W. chinensis* may serve as an important source of bioactive compounds which should be further investigated in animal model for therapeutic potential in the treatment of different chronic diseases.

## 1. Introduction

Medicinal plants have long been used in Bangladesh as folk medicines to treat different ailments. Such plants appear to display diverse biological activities. They contain a variety of chemical constituents that are attributable to their different bioactivities [1,2]. Isolation of compounds with potential bioactivity leads to the development of new drugs for chronic diseases.

*Wedelia chinensis* is a perennial herb of the family Asteraceae and grown in many districts of Bangladesh. The plant has earned reputation for its use in different traditional system of medicine including Ayurveda, Unani, and Siddha. The decoction of the total plant is used to treat the disorders of liver as well as gall bladder [3,4]. The leaves are also used for the management of rheumatic fever, dysentery, and headache. The plant is well known as hair tonic as it promotes the growth of hair [5]. It is also applied for neuroprotective effect [6]. A survey of the literature revealed that the plant has also been used in many Asian countries as traditional medicine. In Taiwan, it is employed as a component of herbal tea and considered to have beneficial effect in relieving cough, fever, intoxication, and hepatic problem [7]. Analysis of chemical constituents demonstrated that the plant contains triterpenoids, diterpenoids, polyphenolics, alkaloids, and steroids [6].

A large number of scientific investigations reported that the plant has different pharmacological activities such as cytotoxic, hepatoprotective, angiogenic, anti-inflammatory, antimicrobial, and antioxidant activities [8,9,10,11,12,13,14]. Although *W. chinensis* displayed a range of potential biological activities, the chemical constituents responsible for the activities are largely unknown. Based on repression of androgen receptor activity in prostrate cancer cells, Lin et al. identified three flavonoid compounds luteolin, wedelolactone, and apigenin for anticancer properties [8]. Aminul et al. also identified the flavonoid apigenin as the active component for acetylcholinesterase and antioxidant activity [15]. All these compounds were found in the aqueous or other polar solvent extracts of the plant.

Motakkin et al. [16] showed earlier that the petroleum ether extract of *W. chinensis* possesses the highest cytotoxic and antimicrobial activities among the different solvent extracts tested. Huang et al. [14] showed the most potent anti-angiogenic activity in the same petroleum ether extract. We also found good antioxidant and cholinesterase inhibitory properties in the petroleum ether extract [15]. These results revealed the petroleum ether extract has diverse biological activity. Investigation of the petroleum ether extract may provide new insights into the bioactive compounds. We, therefore, planned to isolate and characterize the compounds from the petroleum ether extract and determine their cytotoxic, antioxidant, anticholinesterase, and antimicrobial activities.

## 2. Results

### 2.1. Isolation of Compounds

Chromatography of the petroleum ether extract, obtained by direct extraction of the plant powder with petroleum ether, led to the isolation and purification of four compounds. Their structures were determined by analysis of their ^1^H- and ^13^C-NMR spectral data (Appendix A) and a comparison with the values published in the literature [17,18,19,20,21]. They were (-) kaur-16α-hydroxy-19-oic acid (**1**), (-) kaur-16-en-19-oic acid (**2**), β-sitosterol (**3**), and cholesta-5,23-dien-3-ol (**4**) (Figure 1). After identification, all the compounds were studied in details for their cytotoxicity, antioxidant, anticholinesterase, and antimicrobial activities.

### 2.2. Cytotoxic Activity

Brine shrimp lethality bioassay is a simple method used widely for evaluation of the possible cytotoxicity of the plant extracts and compounds [22]. The cytotoxicity of the compounds isolated from the petroleum ether extract was examined by the brine shrimp lethality bioassay and the results are presented in the Figure 2. All the compounds except **3** showed significant cytotoxic effects. The LC_50_ values of the compounds were found to be 12.42 ± 0.87, 16.17 ± 1.11, and 14.27 ± 0.87 µg/mL, respectively, indicating that the compound **1** possesses the highest activity followed by compound **4** and **2**. The standard cytotoxic agent vincristine showed an LC_50_ of 3.48 ± 0.36 µg/mL.

### 2.3. Antioxidant Activity

The antioxidant activity of the compounds was determined by reducing power assay based on the ability to reduce from Fe^3+^ to Fe^2+^ and the results are presented in Figure 3. All the compounds reduced Fe^3+^ in a dose-dependent manner. The compounds **1**–**4** showed an absorbance of 1.55, 1.26, 1.58, and 1.89, respectively at a concentration of 40 µg/mL, while the absorbance of reference standard catechin was 2.14 under the same condition. These results indicated that the steroidal compounds **3** and **4** have relatively higher antioxidant activity than the diterpenoids **1** and **2**.

### 2.4. Anticholinesterase Activity

The anti-AChE and anti-BChE activity of the compounds **1**–**4** was evaluated by Ellman method [23] and the results are presented in the Figure 4. All the four compounds exhibited inhibition of AChE and BChE enzymes. They were able to significantly inhibit both the enzymes at a low concentration of 6.25 μg/mL and the inhibition was found to increase with the increase of the concentration. The diterpene (-) kaur-16-en-19-oic acid (**2**) showed the highest activity against both the enzymes, followed by compound **3**, **4,** and **1**. The IC_50_ values of **2** were 34.82 ± 0.45 µg/mL and 33.70 ± 0.48 µg/mL, respectively. Donepezil, a drug for Alzheimer’s disease (AD) was used as the standard AChE inhibitor which showed an IC_50_ of 7.11 ± 0.24 μg/mL. Similarly, galantamine, which was used as the standard BChE inhibitor, showed an IC_50_ of 8.22 ± 0.11 μg/mL.

### 2.5. Antimicrobial Activity

The antimicrobial potential of the compounds isolated from the petroleum ether extract of *W. chinensis* was screened against five Gram-positive and nine Gram-negative bacteria as well as four fungal species by disc diffusion method and the results are shown in the Table 1 and Table 2. All the compounds except **3** exhibited antibacterial activity against all the test species. The result revealed that the compounds (-) kaur-16α-hydroxy-19-oic acid (**1**) and (-) kaur-16-en-19-oic acid (**2**) have almost similar antibacterial activity and produced a zone of inhibition ranging from 8 mm to 23 mm. Importantly, compound **1** showed strong inhibition against *Staphylococcus aureus* having zone of inhibition of 23 cm. The steroidal compound cholesta-5,23-dien-3-ol (**4**) appeared to be less active than the other two diterpenoids. The standard antibiotic kanamycin exhibited the highest inhibition against the tested bacteria.

In antifungal activity assay, only the two diterpenes were found to exhibit the antifungal activity. While the compound (-) kaur-16-en-19-oic acid (**2**) showed activity against the three fungal species, (-) kaur-16α-hydroxy-19-oic acid (**1)** was inactive against Candida species. The steroidal compounds **3** and **4** did not show any activity against the fungi.

## 3. Discussion

Plant secondary metabolites represent a major source of candidate drugs for different diseases. Screening of plants for active compounds is one of the most important approaches in drug development. It has been reported that a significant number of modern drugs are derived from plants [24]. In recent years, compounds derived from plants have attracted much attention instead of synthetic ones due to safety and multiple pharmacological properties. *W. chinensis* is a potential herb used in different traditional medical systems of Bangladesh to treat various ailments [3,4,5,6]. Until now only several flavonoid compounds have been identified as the major active compounds for anticancer and anticholinesterase activity which are mostly from the polar extract of the plant extract [8,15]. Although numerous compounds have been isolated from the nonpolar extract [14,16], their bioactive potentials are yet to be established. In this paper, we describe the isolation of four compounds from the petroleum ether extract of the plant. The compounds were identified as (-) kaur-16α-hydroxy-19-oic acid (**1**), (-) kaur-16-en-19-oic acid (**2**), β-sitosterol (**3**), and cholesta-5,23-dien-3-ol (**4**) (Figure 1). Although the former three compounds are reported earlier from this plant (6,18), this is the first report of isolation of cholesta-5,23-dien-3-ol from this plant. Since the bioactivity of these compounds is not yet completely known, we studied their cytotoxicity, antioxidant, anti-cholinesterase, and antimicrobial activities.

Cancer is a leading cause of mortality and its prevalence is on the rise worldwide [25]. The limited number of drugs and their side effects account in part for the increasing prevalence of cancer and hence demand the development of new drug for its treatment of this disease. Cytotoxic drugs are recommended for treatment of certain cancers as they kill the cells by damaging DNA and microtubule [26]. Brine shrimp lethality bioassay is a widely used method for evaluating the possible cytotoxicity of the plant extract/compound due to its simplicity and economy. This bioassay has shown a good correlation with the human tumor solid cell lines [27]. Mottakin et al. [16] previously showed the highest cytotoxicity of the petroleum ether extract of *W. chinensis* among the extracts tested by brine shrimp lethality bioassay. We also found the similar result from the petroleum ether extract. In this study, we show that all the compounds except **3** have cytotoxic activity (Figure 2) and the highest activity was displayed by the diterpenoid **1**. The cytotoxic activity of (-) kaur-16-en-19-oic acid isolated from *Wedelia prostrate* was reported by Wu et al. in human HepG2 cells [28]. Our results revealed for the first time that (-) kaur-16α-hydroxy-19-oic acid (**1)** has relatively more cytotoxic activity than (-) kaur-16-en-19-oic acid (**2**). These findings warrant further evaluation of these compounds in human cancer lines.

Oxidative stress is implicated in the pathogenesis of several diseases including cancer, diabetes, Alzheimer’s disease, and other neurodegenerative diseases [29]. In these diseases, the overproduction of reactive oxygen species exceeds the antioxidative defense system. The excessively generated reactive oxygen species causes oxidation of protein, lipid, and DNA which play a role in the pathogenesis of the diseases [30]. Antioxidants from plants have been found to effectively inhibit the oxidation and oxidative damage in the oxidative induced diseases [31]. In this study, we found that all the compounds have antioxidant activity in terms of reducing power. The activity of the steroidal compounds **3** and **4** appeared to be higher than that of the diterpenoids **1** and **2** (Figure 3). The antioxidant activity of the diterpenoids from *Ipomoea nil* and β-sitosterol from *Polygonum hydropiper* were reported by Lee et al. [32] and Ayaz et al. [33], respectively. Our results are consistent with the earlier reports and added that the diterpene (-) kaur-16α-hydroxy-19-oic acid (**1)** has higher antioxidant activity than the diterpene (-) kaur-16-en-19-oic acid (**2**).

Acetylcholine is a neurotransmitter found in the synapses which becomes deficient in Alzheimer’s disease (AD). Acetylholinesterase (AChE) and butyrylcholinesterase (BChE) are expressed in neurons that catalyze the hydrolysis of acetylcholine. At present the AD therapy is based on the inhibition of AChE and BChE that lead to the elevation of acetylcholine level at the synapse and enhance the cholinergic neurotransmission [34]. In this study, we observed that all the compounds are capable of inhibiting both the AChE and BChE enzymes. Among the compounds, the diterpene (-) kaur-16-en-19-oic acid (**2**) showed the highest activity (Figure 4). Jung et al. [35] previously reported the AChE and BChE inhibitory activity of (-) kaur-16-en-19-oic acid and several other Kaurane diterpenoids from *Aralia cordata*. Ayaz et al. [33] found the anticholinesterase activity of β-sitosterol from *Polygonum hydropiper*. The activity of (-) kaur-16-en-19-oic acid (**2**) and β-sitosterol (**3**) obtained in this study were in accordance with the earlier reports [33,35]. So far this is the first report of the anticholinesterase activity of (-) kaur-16α-hydroxy-19-oic acid (**1**) and cholesta-5,23-dien-3-ol (**4**). The anticholinesterase potential of Kaurane diterpenoids further suggests that they are an important class of cholinesterase inhibitor that may be used in AD.

Resistance to antimicrobial agents is one of the emerging problems in the globe. Irrational use and disposal of antibiotics and the mutational capacity of the microbes are the major factors of antibiotic resistance and the emergence of multi drug resistance (MDR) strains [36]. Hence it has become necessary to develop new, safe, and effective antimicrobial agents to combat the infectious diseases. Plant has enormous potential to cure bacterial and fungal infection due to the presence of antimicrobial constituents [37]. In this study, we tested the compounds **1**–**4** against five Gram-positive and nine Gram-negative bacteria as well as four fungal species. All the compounds except **3** showed the antibacterial activity against all the tested species (Table 1). The activity of the diterpenes (-) kaur-16α-hydroxy-19-oic acid (**1**) and (-) kaur-16-en-19-oic acid (**2**) was almost similar which appeared to be higher than that of the steroidal compound **4**. In a previous study, Mottakin et al., [16] reported the antibacterial activity of the compound (-) kaur-16-en-19-oic acid from *W. chinensis* which was in close agreement with our result. There were no reports on the antifungal activity of the compounds from this plant. We found that the compound (-) kaur-16-en-19-oic acid (**2**) has activity against all the three fungal species, while (-) kaur-16α-hydroxy-19-oic acid (**1)** was found inactive against Candida species (Table 2). Our results showed for the first time the antimicrobial activity of (-) kaur-16α-hydroxy-19-oic acid (**1**) and suggest that the presence of diterpenoids might be responsible for the high antimicrobial activity of the petroleum ether extract of the plant.

## 4. Materials and Methods

### 4.1. Chemicals and Organisms

Silica gel 60–120, Silica gel GF_254_, catechin, tricholoroacetic acid (TCA), donezepil, galantamine, and Tris-HCl were procured from Merck, Mumbai, India. Vincristine sulfate, dimethyl sulphoxide (DMSO), potassium ferricyanide, ferric chloride, and Folin-Ciocalteau reagent were from BDH chemicals Ltd., Poole, England. Acetylthiocholine and S-butyrylthiocholine and DTNB were purchased from Sigma-Aldrich, Darmstadt, Germany. Nutrient agar media was from Becton, Dickinson, and Company (Sparks, MD, USA). Petroleum ether and methanol were purchased from Duksan Pure Chemicals Company Limited (Ansan-si, Republic of Korea). The Institute of Nutrition and Food Science, University of Dhaka donated the cultures of bacteria and fungi. The other chemicals were of analytical grade.

### 4.2. Plant Materials

The plant sample was collected from Natore, Bangladesh and identified by Dr. AHM Mahbubur Rahman, Department of Botany, Faculty of Biological Science, University of Rajshahi. A voucher specimen (accession no. 370) was maintained at the herbarium of the Department. A photograph of the plant is shown in Figure 5. After washing with water, the plant material was shade dried and ground into a coarse powder.

### 4.3. Extraction

The ground powder (500 g) of *W. chinensis* was soaked in 2.5 L petroleum ether in an amber bottle and kept for 5 days at room temperature with gentle shaking. The petroleum ether extract (PEE) was then filtered first using a cloth and then by Whatman filter paper 1. It was concentrated under reduced pressure at 40 °C by a rotary evaporator to yield a semisolid mass (3.2 g).

### 4.4. Isolation and Characterization of Compounds

The PEE (3.2 g) from *W. chinensis* was subjected to column chromatography in an open column (20 cm × 2.5 cm) with silica gel 60–120 (100 g) as stationery phase and fractionated by eluting sequentially with n-hexane (100%), n-hexane: chloroform (1:19–19:1), ethylacetate (100%), chloroform: ethylacetate (1:19–19:1), ethylacetate (100%), ethylacetate : methanol (1:19–19:1), and methanol (100%) in a gradually increasing order of polarity. A total of 57 fractions were obtained with 20 mL in each fraction. The fractions were examined on thin layer chromatography (TLC) and then the similar fractions were combined together that yielded eight major fractions (Fr A-Fr H). Fr B (0.61 g) were applied to silica gel preparative thin layer chromatography (PTLC, 20 × 20 cm, 0.75 mm) with n-hexane:ethylacetate (10:1) as mobile phase to yield the compound 1 (19 mg). Similarly fractions Fr C (0.55 g) on PTLC yielded the compound **2** (18 mg) and fractions Fr D (0.49) g yielded the compounds **3** (16 mg) and **4** (20 mg).

A Bruker DPX-400 spectrometer at 400 MHz for ^1^H- and at 100 MHz for ^13^C-NMR spectroscopy was used to identify the compounds. The structures of the compounds were determined by analysis of their spectral data and a comparison with the reported values in the paper [17,18,19,20,21].

#### 4.4.1. (-) Kaur-16α Hydroxy-19 Oic Acid (**1**)

^1^H-NMR (400 MHz, CDCl_3_): δ 2.73 (1H, bt like, H-13), 2.16 (2H, bd, *J* = 10.0 Hz, H-3), 2.01 (2H, s, H-15), 1.26 (3H, s, H-17), 1.24 (3H, s, H-18), 1.01 (each 1H, bd, *J* = 5.2 Hz, H-5 and 9), 0.95 (3H, s, H-20).

^13^C NMR (125 MHz, CDCl_3_): δ 183.17 (C-19), 79.53 (C-16), 57.01 (C-5), 56.03 (C-9), 54.68 (C-15), 47.91 (C-13), 46.21 (C-8), 43.94 (C-4), 42.13 (C-7), 40.85 (C-1), 39.95 (C-10), 38.06 (C-3), 37.67 (C-14), 29.23 (C-18), 26.31 (C-12), 22.20 (C-6), 20.69 (C-17), 19.36 (C-2), 18.98 (C-11), 15.89 (C-20).

#### 4.4.2. (-) Kaur-16-en-19- Oic Acid (**2**)

^1^H-NMR (400 MHz, CDCl_3_): δ 4.80 (2H, d, *J* = 29.5 Hz, H-17), 2.66 (1H, bt like, H-13), 2.18 (2H, bd, *J* = 14.0 Hz, H-3), 2.07 (2H, s, H-15), 2.02 (2H, d, *J* = 11.0 Hz, H-14), 1.91 (2H, bd, *J* = 14.0 Hz, H-1), 1.85~1.88 (2H, m, H-6), 1.67 (2H, bd, *J* = 6.0 Hz, H-12), 1.60 (2H, bd, *J* = 6.0 Hz, H-11), 1.47~1.50 (2H, m, H-2), 1.27 (3H, s, H-18), 1.08~1.11 (1H, m like, H-5 and 9), 0.98 (3H, s, H-20), 0.86 (2H, dt, *J* = 6.5 Hz, H-7);

^13^C NMR (125 MHz, CDCl_3_): δ 184.31 (C-19), 155.61 (C-16), 103.00 (C-17), 57.29 (C-5), 55.35 (C-9), 49.22 (C-15), 44.50 (C-8), 44.12 (C-13), 44.03 (C-4), 41.57 (C-7), 40.99 (C-1), 39.99 (C-14), 39.96 (C-10), 38.08 (C-3), 33.42 (C-12), 29.30 (C-18), 22.20 (C-6), 19.48 (C-2), 18.81 (C-11), 15.98 (C-20).

#### 4.4.3. β-Sitosterol (**3**)

^1^H-NMR (400 MHz, CDCl_3_): δ 4.93~5.18 (1H, m, H-6), 3.22 (1H, tdd, *J* = 11.6Hz and 4.0 Hz, H-3), 1.25 (3H, s, H-18), 1.13 (3H, s, H-19), 0.96 (3H, t, *J* = 23.2 Hz, H-29), 0.87 (3H, s, H-27), 0.87 (3H, s, H-26), 0.81(3H, d, *J* = 16.0 Hz, H-21).

^13^C NMR (125 MHz, CDCl_3_): δ 140.90 (C-5), 121.90 (C-6), 72.00 (C-3), 57.00 (C-14), 56.30 (C-17), 50.40 (C-9), 46.10 (C-24), 42.60 (C-13), 42.50 (C-4), 40.00 (C-12), 37.50 (C-1), 36.70 (C-10), 36.40 (C-20), 34.20 (C-22), 31.90 (C-8), 31.90 (C-7), 31.70 (C-2), 29.40 (C-25), 28.50 (C-16), 26.30 (C-23), 24.50 (C-15), 23.30 (C-28), 21.30 (C-11), 20.00 (C-26), 19.60 (C-18), 19.30 (C-27), 19.00 (C-21), 12.20 (C-29), 12.00 (C-19).

#### 4.4.4. Cholesta-5, 23-Dien-3-ol (**4**)

^1^H-NMR (400 MHz, CDCl_3_): δ 5.35 (1H, bd, *J* = 5.0 Hz, H-6), 5.15 (1H, dd, *J* = 15.0 Hz and 8.5 Hz, H-23), 5.02 (1H, dd, *J* = 15.0 Hz and 8.5 Hz, H-24), 3.52 (1H, sextet/m, H-3), 1.02 (3H, d, *J* = 7.0 Hz, H-21), 0.85 (3H, d, *J* = 6.0 Hz, H-27), 0.82 (3H, s, H-19), 0.80 (3H, d, *J* = 7.0 Hz, H-26), 0.70 (3H, s, H-18).

^13^C NMR (100 MHz, CDCl_3_): δ 140.59 (C-5), 138.13 (C-24), 129.18 (C-23), 121.59 (C-6), 71.93 (C-3), 57.09 (C-14), 56.19 (C-17), 51.47 (C-25), 50.43 (C-9), 42.59 (C-4), 42.50 (C-13), 40.74 (C-20), 39.98 (C-12), 37.57 (C-1), 36.82 (C-10), 32.24 (C-7), 32.19 (C-8), 31.98 (C-2), 29.23 (C-16), 25.74 (C-15), 24.72 (C-22), 21.58 (C-27), 21.43 (C-26), 21.43 (C-11), 19.76 (C-19), 19.36 (C-21), 12.46 (C-18).

### 4.5. Cytotoxicity

The cytotoxic activity of the compounds **1**–**4** was evaluated by the brine shrimp lethality bioassay [22]. In brief, shrimp eggs (*Artemia salina* Lech) were subjected to hatching for 36 h in simulated seawater (3.8% NaCl) at a temperature of 28–30 °C and pH 8.4 with constant oxygen supply to produce nauplii (Larvae). Different concentration (5–80 μg/mL) of the compound was prepared by first dissolving 2 mg of each compound in 400 μL DMSO and then by simulated seawater using serial dilution technique. In separate tubes, 5 mL of the seawater was taken containing 20 nauplii and test solutions were added. The number of survivors after 24 h was counted. Similarly, a control group was used which contained all except compound. The cytotoxicity was expressed in terms of LC_50_ (median lethal concentration) value which was computed from the plot of % mortality against log of sample concentration.

### 4.6. Antioxidant Activity

The antioxidant activity of the compounds **1**–**4** from *W. chinensis* was assessed by their reducing ability using spectrophotometric method [38]. A total of 0.5 mL test compound at different concentration was mixed with 1.25 mL 0.2 M phosphate buffer (pH 6.6) and 1.25 mL of 1% potassium ferricyanide and incubated in a water bath at 50 °C for 20 min. Following incubation, 1.25 mL of 10% tricarboxylic acid solution was added to the reaction mixture, vortexed and centrifuged at 1000× *g* for 10 min. The resulting supernatant (2.5 mL), after mixing with ultrapure water (2.5 mL) and 0.1% ferric chloride solution (0.5 mL), was measured in a spectrophotometer in a cuvette for absorbance at 700 nm. Catechin, a previously reported compound from plant was used as a positive control.

### 4.7. Anti-Acetylcholinesterase (Anti-AChE) and Anti-butyrylcholinesterase Activity (Anti-BChE) Assay

The anti-AChE and anti-BChE activity of the compounds **1**–**4** were determined by the colorimetric method of Ellman et al. [23] using synthetic substrates acetylthiocholine iodide and butyrylthiochoilne, respectively. We used the prepared AChE enzyme from mice brain and BChE enzyme from human blood of a volunteer that has been mentioned earlier [15]. Various concentrations of the compound (500 μL) were mixed with the enzyme solution (500 μL) and incubated at 37 °C for 30 min. A reaction mixture was prepared that contained 0.5 mM acetylthiocholine/butyrylthiocholine and 1 mM DTNB in 50 mM sodium phosphate buffer (pH 8.0). To the reaction mixture (3.5 mL), the enzyme solution containing compound was added, incubated, and immediately the absorbance of the solution was determined at 412 nm. Donezepil was used as the positive control for AChE inhibition and galantamine for BChE inhibition. The percent inhibition was calculated by dividing the difference of sample absorbance from control with control absorbance × 100. The IC_50_ value, the concentration required for 50% inhibition of enzyme activity, was computed by plotting the percent inhibition values against test concentrations of each compound.

### 4.8. Antimicrobial Activity

The compounds were assessed for their antimicrobial potential by disc diffusion method using fourteen bacteria and four fungi [39]. Among the fourteen bacteria, five were Gram positive namely, *Bacillus subtilis* (QL-40), *Sarcina lutea* (QL-1660), *Staphylococcus aureus* (ATCC-259233), *Bacillus megaterium* (QL-38), *Streptococcus-ß-haemolyticus* (ATCC-10389) and nine were Gram negative namely, *Shigella dysenteriae* (AL-35587), *Shigella shiga* (ATCC-26107), *Shigella boydii* (AL-17313), *Shigella sonnei* (AJ-8992), *Shigella flexneriae* (AM-36282), *Escherichia coli* (FPFC-1407), *Salmonella typhi* (ATCC-14028), *Klebsiella* species, and *Pseudomonas aeruginosa* (ATCC-27853). The pathogenic fungi were *Aspergillus niger* (ATCC-1204), *Aspergillus flavus* (ATCC-9807), *Candida* species, and *Fusarium* species. A total of 1 mg of each compound was dissolved in methanol to prepare 10 μg per μL solution and the inhibitory activity was performed with 400 μg of compound per disc. In this method, the test bacteria were inoculated in sterilized nutrient agar media, mixed gently, and transferred to a Petri dish under aseptic condition. The disc containing the compound was placed in a Petri dish and incubated at a temperature of 37 °C for 24 h. For negative control, a disc containing only methanol was used. A clear zone of inhibition was indication of antibacterial activity which was measured in millimeter. As standard antibiotic, kanamycin was used. Similarly, fungal species was cultured at 28 °C for 48 h to 72 h using potato dextrose agar (PDA) media and nystatin was used as standard antifungal agent for comparison.

### 4.9. Statistical Analysis

The values of the experiments were expressed as mean ± SD. All the data were analyzed using Graph Pad Prism (version 8.0.1). One way analysis of variance (ANOVA) was employed to estimate the statistical significance (*p*-value < 0.05) between the average values.

## 5. Conclusions

In conclusion, four compounds including two diterpenes and two steroids were isolated and purified from the petroleum ether extract of *Wedelia chinensis* that are contributing to the cytotoxic, anticholinesterase, antioxidant, and antimicrobial activities. The structures of the compounds were established by studies of their spectral data and a comparison with the published values in the literature. So far this is the first report of isolation of the compound cholesta-5,23-dien-3-ol and a detailed evaluation of the bioactivity of the isolated four compounds from this plant. Our findings suggest that *W. chinensis* is an important source of diterpenoid and steroid compounds with multifunctional activity which may represent potential candidates for management of different chronic diseases. More studies are warranted to confirm their therapeutic activity in animal model.

## Figures and Tables

**Figure 1 molecules-28-00793-f001:**
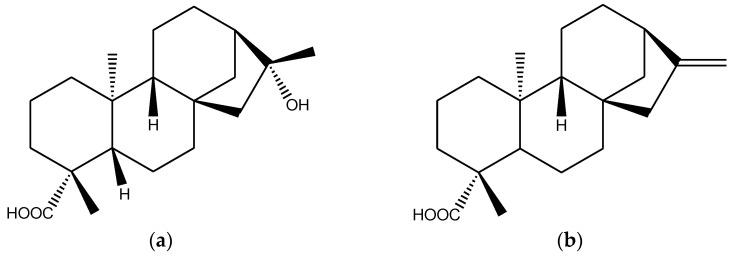
Chemical structures of the compounds **1**–**4**. (**a**), (-) kaur-16α-hydroxy-19-oic acid; (**b**), (-) kaur-16-en-19-oic acid; (**c**), β-sitosterol; (**d**), cholesta-5,23-dien-3-ol.

**Figure 2 molecules-28-00793-f002:**
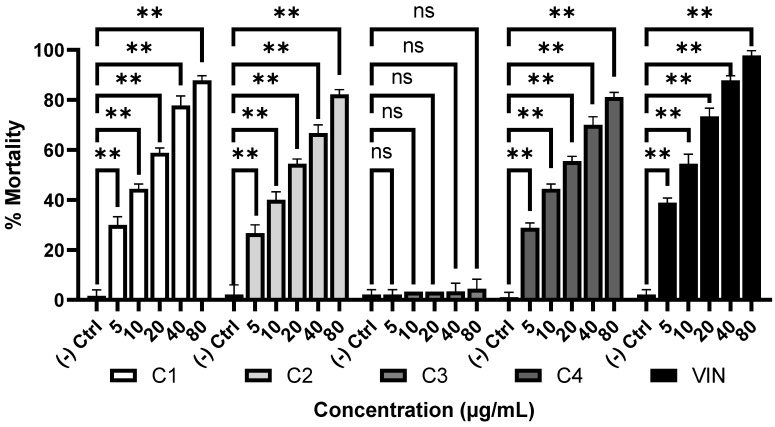
Cytotoxicity of the compounds **1**–**4**. Percent mortality at different concentration of the compounds; LC_50_ (µg/mL). **1**, 12.42 ± 0.87; **2**, 16.17 ± 1.11; **4**, 14.27 ± 0.87; Vincristine, 3.48 ± 0.36. Data are represented as mean ± SD (n = 3). ** *p* < 0.01, significant difference compared with the control group. **1**, (-) kaur-16α-hydroxy-19-oic acid; **2**, (-) kaur-16-en-19-oic acid; **3**, β-sitosterol; **4**, cholesta-5,23-dien-3-ol; VIN, vincristine; ns, not significant.

**Figure 3 molecules-28-00793-f003:**
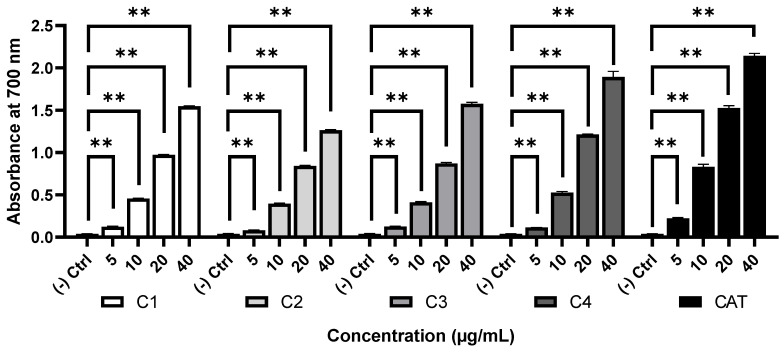
Reducing power of the compounds **1**–**4**. Data are represented as mean ± SD (n = 3). ** *p* < 0.01, significant difference compared with the control group. **1**, (-) kaur-16α-hydroxy-19-oic acid; **2**, (-) kaur-16-en-19-oic acid; **3**, β-sitosterol; **4**, cholesta-5,23-dien-3-ol; CAT, catechin.

**Figure 4 molecules-28-00793-f004:**
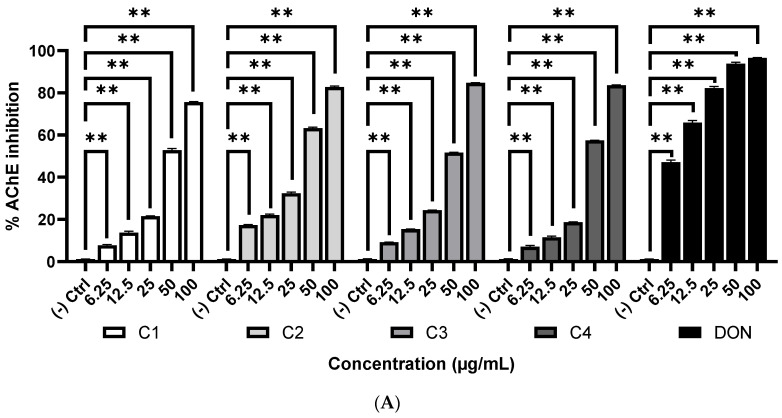
Cholinesterase inhibitory activities of the compounds **1**–**4** and standard. (**A**) Inhibition of acetylcholinesterase (AChE). IC_50_ (µg/mL): **1**, 48.79 ± 0.24; **2**, 34.82 ± 0.45; **3**, 44.49 ± 0.71; **4**, 44.91 ± 0.14; DON, 7.11 ± 0.24. (**B**) Inhibition of butyrylcholinesterase (BChE). IC_50_ (µg/mL): **1**, 46.40 ± 0.35; **2**, 33.70 ± 0.48; **3**, 43.16 ± 0.31; **4**, 47.76 ± 0.41; GAL, 8.22 ± 0.11. Data are represented as mean ± SD (n = 3). ** *p* < 0.01, significant difference compared with the control group. **1**, (-) kaur-16α-hydroxy-19-oic acid; **2**, (-) kaur-16-en-19-oic acid; **3**, β-sitosterol; **4**, cholesta-5,23-dien-3-ol; DON, donepezil; GAL, galantamine.

**Figure 5 molecules-28-00793-f005:**
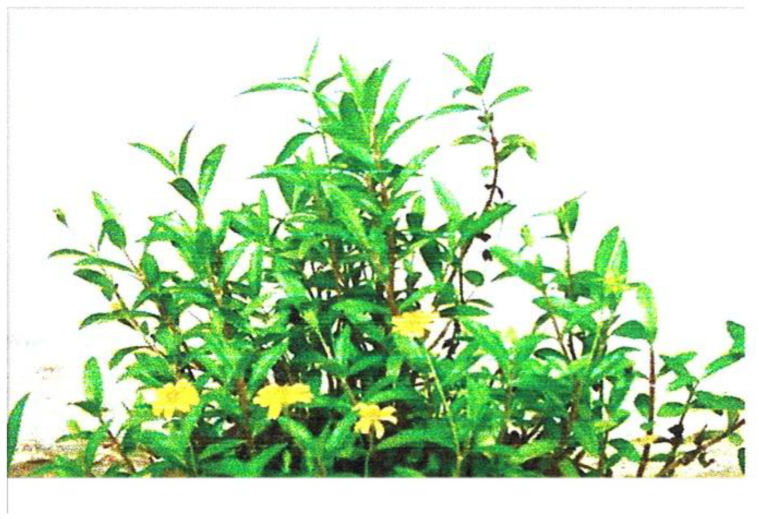
A photograph of the plant Wedelia chinensis.

**Table 1 molecules-28-00793-t001:** Antibacterial activity of the compounds **1**–**4** and standard kanamycin.

Name of Bacteria Strain	Zone of Inhibition (mm)				
	1	2	3	4	Std. Kan
	400 μg/disc				30 μg/disc
Gram Positive					
*1. Bacillus Subtilis*	13	14	−	12	24
*2. Sarcina lutea*	17	16	−	13	29
*3. Staphylococcus aureus*	23	13	−	9	33
*4. Bacillus megaterium*	14	12	−	12	27
*5. Strep-β-haemolyticus*	17	18	−	12	27
Gram Negative					
*1. Shigella dysenteriae*	19	20	−	13	32
*2. Shigella shiga*	14	15	−	14	31
*3. Shigella boydii*	13	14	−	10	31
*4. Shigella sonnei*	15	13	−	12	31
*5. Shigella flexneriae*	12	13	−	11	27
*6. Escherichia coli*	14	16	−	13	27
*7. Salmonella typhi*	13	15	−	10	28
*8. Klebsiella species*	15	17	−	13	28
*9. Pseudomonas aeruginosa*	15	16	−	12	31

Std. Kan indicates standard kanamycin and “−” indicates no activity.

**Table 2 molecules-28-00793-t002:** Antifungal activity of the compounds **1**–**4** and standard nystatin.

	Zone of Inhibition (mm)				
Name of Fungi	1	2	3	4	Std. Nystatin
	400μg/disc				50 μg/disc
1. *Aspergillus niger*	−	−	−	−	29
2. *Aspergillus flavus*	14	17	−	−	30
3. *Candida species*	−	13	−	−	28
4. *Fusarium species*	12	11	−	−	26

Std. indicates standard and “−” indicates no activity.

## Data Availability

The data presented in this study are available on request from the corresponding author.

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
