# Peer review of "Compounds from the Petroleum Ether Extract of Wedelia chinensis with Cytotoxic, Anticholinesterase, Antioxidant, and Antimicrobial Activities"

_molecules, 2023, doi:10.3390/molecules28020793_

Round 1

Reviewer 1 Report

The manuscript by Rashid Khan and colleagues describes the characterization and bioassay of components from a petroleum ether fraction of the traditional medicinal plant Wedelia chinensis. This fraction was previously revealed to have high activity in several assays, so isolation of its phytochemical components is merited. Although it is interesting to read about the bioactivity, these compounds (steroids and diterpenes) and their bioactivities have been reported before, and much of the authors’ work just confirms previous results, so this manuscript is not very novel or innovative. In addition, the experimental section is lacking crucial detail, the statistical analysis requires clarification, and more data should be provided that support the authors’ structural assignments. I do not believe that in its current form this manuscript is appropriate to publish in Molecules. More specific comments are provided below.

1.     For the exception of the antifungal assay results, the data reported in this paper are not of overwhelming novelty, as the discussion mostly says the results for these compounds are consistent with prior reports: see lines 304-305, 316-320, 329-333, and 348-350. Some additional assays should be provided that display novel bioactivity – an unreported cancer cell line, for example, or additional fungi. In addition, the natural products themselves are not terribly unique – sitosterol and kaurenoic acid are widely distributed throughout the plant kingdom and are not unique to this species or accession.

2.     The experimental section for isolation and characterization is not descriptive enough to be repeated by another researcher and needs serious clarification. For example: how much petroleum ether was used for extraction? How many days was this extracted for? (“a couple” is not descriptive – see lines 95-97). The extract was filtered using what media or technique? How much silica gel was used to chromatograph the PEE? What was the ratio of solvents used and how was it increased? What is the size of the fractions that were collected (e.g., are they 1 mL each or 20 mL each?) What is the size or thickness of the preparative TLC plate? Also, the model and field strength of the NMR spectrometer used for analysis should be provided in this section. 

3.     Similarly, the experimental section for the bioassays needs work, or it needs to be stated that these assays are performed exactly as in the literature. Line 159: “potassium buffer” – what type of potassium salt? What pH? What is the volume of the assays? Is this in a microplate or a cuvette, and how much of each reagent is used? Centrifuged for how long and at what speed? Again, for the AChE assays and BChE assays, what volumes of reagents are used? Line 191 reads: “Similarly, the fungus was cultured using potato dextrose agar (PDA) media and nystatin was used as standard antifungal agent for comparison.” What fungus? At what temperature were they grown and for how long? No citation is provided for the conditions.

4.     While a brine shrimp assay is good for assessing general toxicity, a more specific human cell-based assay should be performed here, perhaps using a MTT assay or something similar, especially because the results are compared to a cancer drug and the discussion mentions cancer as a potential use for these agents. Perhaps the authors have a collaborator who can perform these assays?

5.     The characterization of the compounds is not entirely adequate. NMR data from 1H or 13C experiments can be deceiving, especially if a performed at a different field strength or in a different solvent than is used in the literature, and especially for terpene-type compounds, which can have isomers. The identity of these compounds should be confirmed by comparison to an authentic standard that is purchased (for example, beta-sitosterol is commercially available), as well as through mass spectrometry (to confirm a match for mass and fragments). Another NMR experiment such as COSY, HSQC or HMBC should also be used to establish molecular connectivity. The purity of the compounds isolated (such as by HPLC) should also be reported, and all spectra (NMR, etc.) need to be provided as supporting information.

6.     The structures in Figure 1 are poorly drawn (e.g., 3 and 4 have ambiguous stereochemistry) or have low-resolution (1 and 2 were copied from another image). Please re-draw these structures in a program such as ChemDraw (or other) to make them all uniform, and make sure the stereochemistry is correct.

7.     The authors should include a photo of the plant of interest so that readers who are unfamiliar with it can identify it. 

8.     The different letters in Figures 2-4 to indicate statistical significance are confusing. Are the means different from each other? From catechin? It would make much more sense to report significance relative to a negative control (water or DMSO) and statistically compare to the negative control (to prove the compounds are active, with the null hypothesis being that they are not?). Please include a negative control for all of the assays in figures 2-4, the and use standard notation (such as * for P<0.05  or ** for P<0.01) instead of “a, b”, etc.

Specific comments:

Throughout: Italicize genus and species names

Throughout: Bold compound number names

Line 49: Should read “treat disorders of the liver and gallbladder” – “cure” is a very strong word.

Line 48: should read “ayurveda”

Line 50: Should just say “dysentery and headache” – cephalgia is a headache and this is redundant.

Line 59; Should read “cytotoxic”

Line 63: Should read “wedelolactone” – name of natural product is misspelled.

Line 92: Should read “ground into course powder”

Line 169: Should read “butyrylthiocholine”

Line 215: Something for compound 3 should be reported – maybe “no activity at xx concentration”.

Line 274. “plants” should be singular.”

Line 277. Should read “derived from plants”Lines 304-305. This difference (12.42±0.87 vs, 16.17±1.11 ug/mL) is not very significant

Author Response

Point-by-point response to reviewer 1

Reviewer Comment 1: For the exception of the antifungal assay results, the data reported in this paper are not of overwhelming novelty, as the discussion mostly says the results for these compounds are consistent with prior reports: see lines 304-305, 316-320, 329-333, and 348-350. Some additional assays should be provided that display novel bioactivity – an unreported cancer cell line, for example, or additional fungi. In addition, the natural products themselves are not terribly unique – sitosterol and kaurenoic acid are widely distributed throughout the plant kingdom and are not unique to this species or accession.

Author Comment 1: Many thanks for critical analysis of the manuscript. In this study we aimed to isolate and identify the active compounds from the petroleum ether extract of Wedelia chinensis and to determine their roles in bioactivities. According to this objective, we have been able to characterize the four active compounds from the petroleum ether extract of and performed a detailed evaluation of different bioactivities. Of the four isolated compounds, b-sitosterol is most common in plant and the other three compounds are reported to occur in several plants. Although the diterpene (-) kaur-16-en-19-oic acid has been studied for biological activity and only few activities have been known so far, but the bioactivities such as antioxidant, anticholinesterase, cytotoxicity and antimicrobial activities of (-) kaur-16α-hydroxy-19-oic acid and cholesta-5,23-dien-3-ol are not yet known. Therefore, we consider that the findings of this study will provide a comprehensive message regarding the bioactivities of these compounds. Moreover, these evidences must support the traditional uses of this plant.

Reviewer Comment 2.     The experimental section for isolation and characterization is not descriptive enough to be repeated by another researcher and needs serious clarification. For example: how much petroleum ether was used for extraction? How many days was this extracted for? (“a couple” is not descriptive – see lines 95-97). The extract was filtered using what media or technique? How much silica gel was used to chromatograph the PEE? What was the ratio of solvents used and how was it increased? What is the size of the fractions that were collected (e.g., are they 1 mL each or 20 mL each?) What is the size or thickness of the preparative TLC plate? Also, the model and field strength of the NMR spectrometer used for analysis should be provided in this section.

Author Comment 2:  According to the suggestion we have improved the experimental section that has been marked in red color. 

Reviewer Comment 3.   Similarly, the experimental section for the bioassays needs work, or it needs to be stated that these assays are performed exactly as in the literature. Line 159: “potassium buffer” – what type of potassium salt? What pH? What is the volume of the assays? Is this in a microplate or a cuvette, and how much of each reagent is used? Centrifuged for how long and at what speed? Again, for the AChE assays and BChE assays, what volumes of reagents are used? Line 191 reads: “Similarly, the fungus was cultured using potato dextrose agar (PDA) media and nystatin was used as standard antifungal agent for comparison.” What fungus? At what temperature were they grown and for how long? No citation is provided for the conditions. 

Author Comment 3:  In line with the reviewer comment, we have described the experimental section of bioassay.

Reviewer Comment 4. While a brine shrimp assay is good for assessing general toxicity, a more specific human cell-based assay should be performed here, perhaps using a MTT assay or something similar, especially because the results are compared to a cancer drug and the discussion mentions cancer as a potential use for these agents. Perhaps the authors have a collaborator who can perform these assays? 

Author Comment 4: Since brine shrimp bioassay has been used earlier for evaluation of cytotoxicity of the petroleum ether extract of the plant, we therefore used the same bioassay to explore the active compounds. Due to limitation at the present moment it is not possible to perform cell based bioassay, but expect to be done in the future. However we appreciate the reviewer for nice suggestion.

Reviewer Comment 5.  The characterization of the compounds is not entirely adequate. NMR data from 1H or 13C experiments can be deceiving, especially if a performed at a different field strength or in a different solvent than is used in the literature, and especially for terpene-type compounds, which can have isomers. The identity of these compounds should be confirmed by comparison to an authentic standard that is purchased (for example, beta-sitosterol is commercially available), as well as through mass spectrometry (to confirm a match for mass and fragments). Another NMR experiment such as COSY, HSQC or HMBC should also be used to establish molecular connectivity. The purity of the compounds isolated (such as by HPLC) should also be reported, and all spectra (NMR, etc.) need to be provided as supporting information. 

Author Comment 5: According to the comment, we have added supplementary data (Figure S1-S11) to support the structure of the compounds.

Reviewer Comment 6.  The structures in Figure 1 are poorly drawn (e.g., 3 and 4 have ambiguous stereochemistry) or have low-resolution (1 and 2 were copied from another image). Please re-draw these structures in a program such as ChemDraw (or other) to make them all uniform, and make sure the stereochemistry is correct.

Author Comment 6: The structure of the compounds was redrawn by using ChemDraw.

Reviewer Comment 7. The authors should include a photo of the plant of interest so that readers who are unfamiliar with it can identify it.

Author Comment 7:  A photograph of the plant (Figure 1) has been included in the manuscript.

Reviewer Comment 8.  The different letters in Figures 2-4 to indicate statistical significance are confusing. Are the means different from each other? From catechin? It would make much more sense to report significance relative to a negative control (water or DMSO) and statistically compare to the negative control (to prove the compounds are active, with the null hypothesis being that they are not?). Please include a negative control for all of the assays in figures 2-4, the and use standard notation (such as * for P<0.05  or ** for P<0.01) instead of “a, b”, etc.

Author Comment 8: According to suggestion, we have improved the statistical analysis and hope it would be OK.

Reviewer Comment 9.  Specific comments:

Throughout: Italicize genus and species names

Throughout: Bold compound number names

Line 49: Should read “treat disorders of the liver and gallbladder” – “cure” is a very strong word.

Line 48: should read “ayurveda”

Line 50: Should just say “dysentery and headache” – cephalgia is a headache and this is redundant.

Line 59; Should read “cytotoxic”

Line 63: Should read “wedelolactone” – name of natural product is misspelled.

Line 92: Should read “ground into course powder”

Line 169: Should read “butyrylthiocholine”

Line 215: Something for compound 3 should be reported – maybe “no activity at xx concentration”.

Line 274. “plants” should be singular.”

Line 277. Should read “derived from plants”Lines 304-305. This difference (12.42±0.87 vs, 16.17±1.11 ug/mL) is not very significant

Author Comment 9:  We have corrected all according to the specific comments.

Reviewer 2 Report

Here are some suggestions for revised-manuscript:

1)    The method of isolation and characterization of compounds should be re-write in a better way and with more details.

-     In lines 100-101, the ratios of the mobile phase solvents should clearly be mentioned.

-     I think 57 fractions are very much, so it is better to reduce them to 10 fractions by combining similar fractions.

-     In the spectroscopic analysis of the compounds, please use DEPT NMR to confirm the type of carbon (q, d, t or s).

2)    As five concentrations of the compounds have been tested, there is no compatibility between the figures and the text. The author should clearly state the activity values in the text related to which concentration.

3)    In cytotoxicity activity, the author should provide the result (LC50 value)of compound 3 in the text and figure 2, even if there is no significant effect.

4)   In anticholinesterase activity, lines 243-244, there is only one IC50value for Donepezil, although there should be two IC50 values for the inhibition of two enzymes AChE and BChE.

Author Response

Point-by-point response to reviewer 2

Reviewer Comment 1.    The method of isolation and characterization of compounds should be re-write in a better way and with more details.

-     In lines 100-101, the ratios of the mobile phase solvents should clearly be mentioned.

-     I think 57 fractions are very much, so it is better to reduce them to 10 fractions by combining similar fractions.

-     In the spectroscopic analysis of the compounds, please use DEPT NMR to confirm the type of carbon (q, d, t or s).

Author Comment 1: According to the suggestion we have improved the experimental section as marked in red color and added DEPT data of the compounds.

Reviewer Comment 2:  As five concentrations of the compounds have been tested, there is no compatibility between the figures and the text. The author should clearly state the activity values in the text related to which concentration.

Author Comment 2: Since IC50 or LC50 values, half maximal inhibitory concentration or half maximal lethal concentration are most suitable for comparison of the activities of the test compounds, we therefore used these values rather than a particular concentration.

Reviewer Comment 3: In cytotoxicity activity, the author should provide the result (LC50 value) of compound 3 in the text and figure 2, even if there is no significant effect.

Author Comment 3: We have done accordingly.

Reviewer Comment 4: In anticholinesterase activity, lines 243-244, there is only one IC50value for Donepezil, although there should be two IC50 values for the inhibition of two enzymes AChE and BChE.

Author Comment 4: Accordingly we have mentioned the IC50 value of galantamine which was used as a standard BChE.

Reviewer 3 Report

Manuscript ID: molecules-2059259, titled “Compounds from the petroleum ether extract of Wedelia chinensis with cytotoxic, anticholinesterase, antioxidant and antimicrobial activities” is well-structured. It states the purpose of the research, so I have a few following comments for the authors regarding their study.

Pay attention to compound nouns throughout text, somewhere they are written with/without dash, check all of them (lines: 32, 60, 75, 206, 360...); should be uniformly (according the title).

All abbreviations should be defined on their first mention in the text (e.g. PEE (line 97); TCA (line 160); AD (line 243)).

Concerning Antimicrobial activity in Materials and Methods part please define which strains of microorganisms were used? It will make your experiments more reproducible.

Take care about cite literatures, usually it is in brackets (see line 203).

It would be fine to supplement Figure 2 with finding that compound 3 had no cytotoxic activity.

For most spectrometers and colorimeters, the useful absorbance range is from 0.1 to 1. Absorbance values greater than or equal to 1.0 are too high. If you are getting absorbance values of 1.0 or above, your solution is too concentrated. Hence, suggestion is to dilute your samples and recollect data regarding antioxidant activity.

In lines 269-270 the authors said “activity against all the three fungal species”, four fungi strains were analyzed, right?

In order to improve your discussion, you could make comparison between previously assessed petroleum ether extract of Wedelia chinensis concerning cytotoxicity (line 299) and such individual compounds potential from your study.

Author Response

Point-by-point response to reviewer 3

Reviewer Comment 1: Pay attention to compound nouns throughout text, somewhere they are written with/without dash, check all of them (lines: 32, 60, 75, 206, 360...); should be uniformly (according the title).

Author Comment 1: We have corrected all throughout the manuscript.

Reviewer Comment 2: All abbreviations should be defined on their first mention in the text (e.g. PEE (line 97); TCA (line 160); AD (line 243)).

Author Comment 2: We have done accordingly.

Reviewer Comment 3: Concerning Antimicrobial activity in Materials and Methods part please define which strains of microorganisms were used? It will make your experiments more reproducible.

Author Comment 3: Very good suggestion. We have mentioned the strain of microorganisms.

Reviewer Comment 4: Take care about cite literatures, usually it is in brackets (see line 203).

Author Comment 4: We have corrected the mistake.

Reviewer Comment 5: It would be fine to supplement Figure 2 with finding that compound 3 had no cytotoxic activity.

Author Comment 5: Accordingly we have done.

Reviewer Comment 6: For most spectrometers and colorimeters, the useful absorbance range is from 0.1 to 1. Absorbance values greater than or equal to 1.0 are too high. If you are getting absorbance values of 1.0 or above, your solution is too concentrated. Hence, suggestion is to dilute your samples and recollect data regarding antioxidant activity.

Author Comment 6: In reducing power assay, we found that Beer’s law is followed at the concentration that gave the absorbance values in the range between 0.1 to 2.0 or little above. According to suggestion, we have rearranged the figure.

Reviewer Comment 7: In lines 269-270 the authors said “activity against all the three fungal species”, four fungi strains were analyzed, right?

Author Comment 7: We have corrected the mistake.

Reviewer Comment 8: In order to improve your discussion, you could make comparison between previously assessed petroleum ether extract of Wedelia chinensis concerning cytotoxicity (line 299) and such individual compounds potential from your study.

Author Comment 8: Accordingly we have tried to make a comparison.

Round 2

Reviewer 1 Report

 The re-submission of the manuscript is much improved (thanks to the authors for their corrections), but still could use some substantial revisions before publication.

1.     The reviewer is sensitive to the fact that English may not be the authors’ native language, but this manuscript could use some proofreading and correction of the English style and grammar. 

2.     In the introduction, the authors state “It is also applied to use in strengthening nervous system (6)”. This is a vague description - What does this mean? Is it neuroprotective? This should be clarified or removed.

3.     Are these four isolated components the major components of the petroleum ether extract? An HPLC or LC-MS chromatogram should be collected (and each component should be indicated in the chromatogram and shown in a figure), and some comment on this should be made. Otherwise, it is possible that the compounds isolated are neither the most active the most abundant, or not responsible for the extract’s activity.

4.     Additional details are needed about the brine shrimp assay. At what temperature were the eggs incubated? Was this under air or was a special atmosphere? What % salinity?

5.     Upon examination of the NMR spectra in the supplemental data, the reviewer notes some minor impurities in the 1H NMR spectrum. Some measure of purity of these compounds should be provided, such as a HPLC chromatogram or LC-MS chromatogram, the data should be included in the supplemental material, and the purity should be commented on in the manuscript.

6.     In Figure S7, there is a gap in the spectrum baseline. Please show the full spectrum and do not truncate or splice spectra. 

7.     Figure 1: the structures of compounds 3 and 4 are still not chemically correct. Please put the correct stereochemistry into these two drawings. The right-hand side of molecule 4 should not have two chiral methyl groups – this is not a chiral center.

8.     Specific comments: 

-Section 2.2. Should read “a photograph of the plant” (photograph is misspelled.)

-Line 186: “was subjected to a spectrophotometer” should read “was measured in a spectrophotometer”

-Line 202: “control absorbance X 100” - should use a lower-case x or the times/cross-product symbol. 

-Line 361 "Butyrylcholinesterase" is misspelled.

Author Response

Reviewer Comment 1. The reviewer is sensitive to the fact that English may not be the authors’ native language, but this manuscript could use some proofreading and correction of the English style and grammar. 

Author Comment 1: The manuscript has been reviewed by an expert and hope the manuscript will be accepted in this form.

Reviewer Comment 2.     In the introduction, the authors state “It is also applied to use in strengthening nervous system (6)”. This is a vague description - What does this mean? Is it neuroprotective? This should be clarified or removed.

Author Comment 2: We have corrected according to the comments of the reviewer.

Reviewer Comment 3.     Are these four isolated components the major components of the petroleum ether extract? An HPLC or LC-MS chromatogram should be collected (and each component should be indicated in the chromatogram and shown in a figure), and some comment on this should be made. Otherwise, it is possible that the compounds isolated are neither the most active the most abundant, or not responsible for the extract’s activity.

Author Comment 3: It is obvious that all the isolated four compounds are major compounds. It may be mentioned that our main aim was to isolate and identify the compounds from the petroleum ether fraction and to assess their role in biological activity. For isolation of compounds, the petroleum ether fraction was first fractionated by column chromatography with silica gel as stationery phase and different solvent ratio with increasing polarity as eluents as have been mentioned in the Material and Method. All the fractions were checked on TLC and the fractions with similar profile were combined together (fr A- Fr H) which was then measured. After TLC analysis, the compounds in the Fr B, Fr C and Fr D were targeted due to their relative abundance in the fraction. Fr B, Fr C and Fr D were further purified by PTLC that afforded four compounds. So they are apparently considered as the major compounds. Due to limitation of the available facilities at that time, we were not in a position to analyze the compounds or fraction by HPLC or LC/MS. However, we have plan in the future to determine the quantity of the active compounds in the plant. Thanks for your nice suggestion.

Reviewer Comment 4.     Additional details are needed about the brine shrimp assay. At what temperature were the eggs incubated? Was this under air or was a special atmosphere? What % salinity?

Authors Comment 4: We have provided additional details about the brine shrimp assay. We have addressed all the points.

Reviewer Comment 5.     Upon examination of the NMR spectra in the supplemental data, the reviewer notes some minor impurities in the 1H NMR spectrum. Some measure of purity of these compounds should be provided, such as a HPLC chromatogram or LC-MS chromatogram, the data should be included in the supplemental material, and the purity should be commented on in the manuscript.

Author Comment 5: It is usual that the compounds, which are pure, are selected for analysis by NMR spectra to establish the structure. Because the impurities, if present, in the pure compounds is reflected in the 1H and 13C NMR spectra by appearing signals which seriously interferes with the analysis and makes difficult to elucidate the structure. In our case, we made sure on TLC analysis that the compounds are pure and thus we have been able to establish the structure of the compounds by analysis of the spectra. We regret to let you know you that due to limitation of the available facilities at that time, we were not in a position to analyze the compounds or fraction by HPLC or LC/MS.

Reviewer Comment 6.     In Figure S7, there is a gap in the spectrum baseline. Please show the full spectrum and do not truncate or splice spectra. 

Authors Comment 6: It was simply a printing problem and we hope the current spectrum is OK. We took almost same scale for each spectrum which was required.

Reviewer Comment 7.     Figure 1: the structures of compounds 3 and 4 are still not chemically correct. Please put the correct stereochemistry into these two drawings. The right-hand side of molecule 4 should not have two chiral methyl groups – this is not a chiral center.

Authors Comment 7: We are very sorry for the mistake. We have corrected the structures.

Reviewer Comment 8.     Specific comments: 

-Section 2.2. Should read “a photograph of the plant” (photograph is misspelled.)

-Line 186: “was subjected to a spectrophotometer” should read “was measured in a spectrophotometer”

-Line 202: “control absorbance X 100” - should use a lower-case x or the times/cross-product symbol. 

-Line 361 "Butyrylcholinesterase" is misspelled.
